# CONVERGENCE OF THE MINI-BATCH SIHT ALGORITHM

## ABSTRACT

The Iterative Hard Thresholding (IHT) algorithm has been considered extensively as an effective deterministic algorithm for solving sparse optimizations. The IHT algorithm benefits from the information of the batch (full) gradient at each point and this information is a crucial key for the convergence analysis of the generated sequence. However, this strength becomes a weakness when it comes to machine learning and high dimensional statistical applications because calculating the batch gradient at each iteration is computationally expensive or impractical. Fortunately, in these applications the objective function has a summation structure that can be taken advantage of to approximate the batch gradient by the stochastic mini-batch gradient. In this paper, we study the mini-batch Stochastic IHT (SIHT) algorithm for solving the sparse optimizations. As opposed to previous works where increasing and variable mini-batch size is necessary for derivation, we fix the mini-batch size according to a lower bound that we derive and show our work. To prove stochastic convergence of the objective value function we first establish a critical sparse stochastic gradient descent property. Using this stochastic gradient descent property we show that the sequence generated by the stochastic mini-batch SIHT is a supermartingale sequence and converges with probability one. Unlike previous work we do not assume the function to be a restricted strongly convex. To the best of our knowledge, in the regime of sparse optimization, this is the first time in the literature that it is shown that the sequence of the stochastic function values converges with probability one by fixing the mini-batch size for all steps.

## 1 INTRODUCTION

We consider the following sparse optimization problem:

$$\text{(P)}: \quad \begin{array}{l} \min f(\mathbf{x}, \Xi) := \frac{1}{N} \sum_{i=1}^{N} f^{(i)}(\mathbf{x}, \xi^{(i)}) \\ \text{s.t. } \mathbf{x} \in C_s \end{array} \tag{1}$$

where $f^{(i)} : \mathbb{R}^n \times \Xi \to \mathbb{R}$ for $i = 1, \dots, N$, $\Xi = \{\xi^{(1)}, \dots, \xi^{(N)}\}$, and $C_s = \{\mathbf{x} \in \mathbb{R}^n \mid \|\mathbf{x}\|_0 \leq s\}$ (sparsity constraint) is the union of finitely many subspaces whose dimension is less than or equal to the sparsity level $s$ such that $1 \leq s < n$. The importance of the Problem (P) is due to the fact that finding a sparse network whose accuracy is on a par with a dense network amounts to solving a bi-level, constrained, stochastic, nonconvex, and non-smooth sparse optimization problem Damadi et al. (2022). Thus finding efficient algorithms that solve Problem (P) can be beneficial for addressing compression of deep neural networks.

Among algorithms for solving sparse optimization the Iterative Hard Thresholding (IHT) algorithm has been a very successful one due to the simplicity of its implementation. The IHT algorithm not only has been practically efficient, but also shows theoretical promising results. It was originally devised for solving compressed sensing problems in 2008 Blumensath & Davies (2008; 2009). Since then, a large body of literature has been studying it from different perspectives. For example, Beck & Eldar (2013); Lu (2014; 2015); Pan et al. (2017); Zhou et al. (2021) consider convergence of iterations, Jain et al. (2014); Liu & Foygel Barber (2020) study the limit of the objective function value sequence, Liu et al. (2017); Zhu et al. (2018) address duality, Zhou et al. (2020); Zhao et al. (2021) extend it to Newton's-type IHT, Blumensath (2012); Khanna & Kyrillidis (2018); Vu & Raich (2019); Wu & Bian (2020) address accelerated IHT, and Wang et al. (2019); Bahmani et al. (2013) solve logistic regression problem using the IHT algorithm. Recently Damadi & Shen (2022) introduced the concepts of HT-unstable stationary points (saddle points in the sense of sparse optimization) and

---

**Algorithm 1** The mini-batch stochastic iterative hard thresholding

---

**Require:** $\mathbf{x}^0 \in C_s$ such that $\|\mathbf{x}^0\|_0 \leq s$, a stepsize $0 < \gamma < \frac{1}{L_s}$, and $1 \leq S_B \in \mathbb{N}$ such that

$$S_B \geq \frac{N}{1 + \frac{1 - L_s \gamma}{1 + L_s \gamma} \frac{N-1}{\frac{c}{N} - 1}}$$

for some $c > 0$.
1: **for** $k = 0, 1, \ldots$ **do**
2:     Construct $B^k$ by selecting $S_B$ elements from $\{1, \ldots, N\}$ uniformly without replacement such that $|B^k| = S_B$.
3:     Calculate the stochastic mini-batch gradient as $\mathcal{G}(\mathbb{X}^k, \Xi, B^k) = \frac{1}{S_B} \sum_{i \in B^k} \nabla f^{(i)}(\mathbb{X}^k, \xi^{(i)})$.
4:     $\mathbb{X}^{k+1} \in H_s(\mathbb{X}^k - \gamma \mathcal{G}(\mathbb{X}^k, \Xi, B^k))$.

---

showed the escapability property of the HT-unstable stationary points as one of the crucial properties of the IHT algorithm. Also, they showed Q-linearly convergence of the IHT algorithm towards strictly HT-stable stationary points. However, these desirable properties, requires to compute the batch (full) gradient at each iteration which is computationally expensive or impractical with current GPUs.

On the other hand, almost all training for deep neural networks are done using the mini-batch stochastic gradient which is a combination of the stochastic approximation Robbins & Monro (1951) implemented by the backpropagation algorithm Rumelhart et al. (1986). By taking the mini-batch stochastic approximation, we consider solving Problem (P) using the mini-batch Stochastic Iterative Hard Thresholding algorithm outlined in Algorithm 1. Similar to practice where the mini-batch size is fixed beforehand, we fix the mini-batch size at the beginning which is different from previous work Zhou et al. (2018) in this area. Also, for showing our theoretical results we directly use the mini-batch stochastic gradient and derive our theoretical results which is different from previous works Chen & Gu (2016); Li et al. (2016) where the batch (full) gradient is used to show the theoretical results. As opposed to other works where restricted strong convexity is necessary for deriving convergence results Liang et al. (2020); Zhou et al. (2018), here the only assumption we make is the restricted strong smoothness on the objective function not on each individual one. Also, we assume that the objective function is a bounded below function which is the case for objective functions used in machine learning applications. Similar to practice where the mini-batch size is fixed beforehand, we fix the mini-batch size at the beginning which is different from previous works Zhou et al. (2018).

SUMMARY OF CONTRIBUTIONS

By considering the mini-batch SIHT Algorithm 1 for Problem (P), we develop the following results:

- We establish a new critical sparse stochastic gradient descent property of the hard thresholding (HT) operator that has not been found in the literature.

- For a given step-size $0 < \gamma < \frac{1}{L_s}$, we find a lower bound on the size of the mini-batch that guarantees the expected descent of the objective value function after hardthresholding.

- Using the sparse stochastic gradient descent property we show that the sequence generated by the mini-batch SIHT algorithm is supermartingale and converges with probability one.

- We show that for a certain class of functions in Problem (P) where $f(\mathbf{x}, \xi^i) := f^{(i)}(\mathbf{V}_{i\bullet}\mathbf{x})$ $f^{(i)} : \mathbb{R}^n \to \mathbb{R}$, the sum of norm squared of individual gradients restricted to a set of some elements $\mathcal{J}$, i.e., $\sum_{i=1}^N \|\nabla_{\mathcal{J}} f^{(i)}\|_2^2$, evaluated at every point is proportionate to the norm of the batch gradient $\|\nabla_{\mathcal{J}} f\|_2^2$ where the proportionality constant only depends on the data. Moreover, dependency of the proportionality constant on the data is restricted to the set of $\mathcal{J}$ not the entire data.

## 2   RELATED WORK

In order improve computational efficiency of the IHT algorithm, algorithms based on stochastic hard thresholding try to use the finite-sum structure of problem (P) Nguyen et al. (2017); Li et al.

(2016); Shen & Li (2017). The StoIHT algorithm is introduced in Nguyen et al. (2017) where at each iteration a random element from the sum in Problem (P) is drawn and the associated gradient is calculated. Basically, the gradient is approximated by a mini-batch stochastic gradient with size one. The StoIHT algorithm defines a sparse subspace and then projects the updated vector into that. To show the theoretical results in Nguyen et al. (2017), the restricted strong smoothness condition for each individual function in Problem (P) is required as well as the restricted strong convexity for the objective function. In addition, the StoIHT algorithm needs the restricted condition number be to 4/3 which is hard to meet in practice. The stochastic variance reduced gradient hard thresholding (SVRG-HT) algorithm Li et al. (2016); Shen & Li (2017) tries to mitigate the variance with a cost of calculating the (batch) full gradient at some stages. This information of the batch gradient is the key for reducing the variance. Similar to the StoIHT algorithm, the SVRG-HT algorithm requires the restricted strong smoothness condition for each individual function in Problem (P) as well as the restricted strong convexity for the objective function. The Accelerated Stochastic Block Coordinate Gradient Descent with Hard Thresholding (ASBCDHT) algorithm in Chen & Gu (2016) is a randomized version of the StoIHT algorithm which suffers the drawbacks of the StoIHT algorithm, i.e., calculating the full gradient and requirement of the restricted strong conditions. The Hybrid Stochastic Gradient Hard Thresholding (HSG-HT) algorithm in Zhou et al. (2018) is a variant of stochastic IHT algorithms that uses a mini-batch stochastic gradient at each step. However, from the theoretical perspective, the size of a mini-batch has to increase as the algorithm progresses. This makes the algorithm almost deterministic in calculating the gradient and defeats the purpose of using the mini-batch stochastic gradient. The stochastically controlled stochastic gradients (SCSG-HT) algorithm in Liang et al. (2020) uses mini-batch stochastic gradients with large batch size as opposed to the SVRG-HT and the ASBCDHT algorithms to reduce the variance with less computation, i.e., not calculating the batch gradient at some steps.

We present the mini-batch stochastic IHT algorithm and show that the stochastic sequence of the function value is a supermartingale sequence and it converges with probability one. To show our result, we assume the objective function has the restricted strong smoothness property and is bounded below which is the case for objective functions used machine learning applications. Also, to the best of our knowledge, in the regime of sparse optimization, this is the first time in the literature that it is shown that the sequence of the stochastic function values converges with probability one by fixing the mini-batch size for all steps.

## 3 DEFINITIONS

We provide some definitions that will be used throughout the paper.

**Definition 1** (Restricted Strong Smoothness (RSS)). *A differentiable function $f : \mathbb{R}^n \to \mathbb{R}$ is said to be restricted strongly smooth with modulus $L_s > 0$ or is $L_s$-RSS if*

$$f(\mathbf{y}) \leq f(\mathbf{x}) + \langle \nabla f(\mathbf{x}), \mathbf{y} - \mathbf{x} \rangle + \frac{L_s}{2} \|\mathbf{y} - \mathbf{x}\|_2^2 \quad \forall \mathbf{x}, \mathbf{y} \in \mathbb{R}^n \text{ such that } \|\mathbf{x}\|_0 \leq s, \|\mathbf{y}\|_0 \leq s. \quad (2)$$

**Definition 2** (The HT operator). *The HT operator $H_s(\cdot)$ denotes the orthogonal projection onto multiple subspaces of $\mathbb{R}^n$ with dimension $1 \leq s < n$, that is,*

$$H_s(\mathbf{x}) \in \arg \min_{\|\mathbf{z}\|_0 \leq s} \|\mathbf{z} - \mathbf{x}\|_2. \quad (3)$$

**Claim 1.** *The HT operator keeps the $s$ largest entries of its input in absolute values.*

For a vector $\mathbf{x} \in \mathbb{R}^n$, $\mathcal{I}_s^{\mathbf{x}} \subset \{1, \dots, n\}$ denotes the set of indices corresponding to the first $s$ largest elements of $\mathbf{x}$ in absolute values. For example $H_2([1, -3, 1]^\top)$ is either $[0, -3, 1]^\top$ or $[1, -3, 0]^\top$ where $\mathcal{I}_2^{\mathbf{y}} = \{2, 3\}$ and $\mathcal{I}_2^{\mathbf{y}} = \{1, 2\}$, respectively. Therefore, the output of it may not be unique. This clearly shows why HTO is not a convex operator and why there is an inclusion in (3) not an inequality.

**Definition 3** (Convergence with probability one). *A random sequence $(\mathbf{x}^k \in \mathbb{R}^n)$ in a sample space $\Omega$ converges to a random variable $\mathbf{x}^*$ with probability one if*

$$\mathbb{P}\Big[\omega \in \Omega : \lim_{k \to \infty} \|\mathbf{x}^k(\omega) - \mathbf{x}^*\|\Big] = 0.$$

# 4 RESULTS

We consider solving Problem (1) using the mini-batch SIHT Algorithm 1 and develop results that guarantee the convergence of the sequence of function values generated by the SIHT Algorithm. To do so, we present our results in two separate subsections. The first part provides stochastic results characterizing expectation of functions involving the sample average of given vectors. Then, in the subsequent subsection we use the aforementioned results to show Theorem 3 which establishes a stochastic gradient result that is the foundation for the convergence of the function value sequence.

## 4.1 STOCHASTIC RESULTS FOR SAMPLE AVERAGE

In this subsection, we consider a sample average whose elements are drawn uniformly and without replacement. Then, we prove Lemma 2 that calculates the expected value of the norm squared of the sample average based on the covariance matrix of a random vector whose elements are Bernoulli random variable determining elements of the sample average. Next, in Corollary 1 using Lemma 2 we calculate the expected value of the squared distance between the sample average and the overall average. This result is extended in Theorem 1 where the expected value is calculated so that one is able to find the mentioned expectation based on each individual vector and the overall average. We start with the following well-known lemma.

**Lemma 1** (Mathai & Provost (1992)). *Let $\mathbf{\Lambda} \in \mathbb{R}^{n \times n}$ be a deterministic matrix and $\boldsymbol{\xi} \in \mathbb{R}^n$ be a random vector that is distributed according to some probability distribution $\mathcal{P}$. Then,*

$$\mathbb{E}_{\boldsymbol{\xi}}\left[\boldsymbol{\xi}^\top \mathbf{\Lambda} \boldsymbol{\xi}\right] = trace(\mathbf{\Lambda} Cov(\boldsymbol{\xi})) + \mathbb{E}_{\boldsymbol{\xi}}^\top\left[\boldsymbol{\xi}\right] \mathbf{\Lambda} \mathbb{E}_{\boldsymbol{\xi}}\left[\boldsymbol{\xi}\right]. \tag{4}$$

To invoke the above lemma, notice that one can define a random vector whose elements are Bernoulli random variables determining whether the associated vector is in the sample average or not. Thus we prove the following lemma.

**Lemma 2.** *Let $\mathbf{g}^{(1)}, \ldots, \mathbf{g}^{(N)} \in \mathbb{R}^n$ be $N$ deterministic vectors and $B \subseteq \{1, \ldots, N\}$ be a random set. Let $\bar{\mathbf{g}} := \frac{1}{N}\sum_{i=1}^{N} \mathbf{g}^{(i)}$, $\mathcal{G}(B) := \frac{1}{|B|}\sum_{i \in B} \mathbf{g}^{(i)}$, $\mathbf{G} := \begin{bmatrix} \mathbf{g}^{(1)} & \cdots & \mathbf{g}^{(N)} \end{bmatrix} \in \mathbb{R}^{n \times N}$, and $\mathbf{z}(B) = [z_1(B), \ldots, z_N(B)]^\top$ where $z_i(B)$ is a Bernoulli random variable such that $z_i(B) = 1$ if $i \in B$ otherwise $z_i(B) = 0$ for $i = 1, \ldots, N$. Assume $\mathbb{E}_B[\mathcal{G}(B)] = \bar{\mathbf{g}}$, then for any random set $B$ with fixed size $|B|$, the following holds:*

$$\mathbb{E}_B\left[\|\mathcal{G}(B)\|^2\right] = \frac{1}{|B|^2} trace\left(\mathbf{G}^\top \mathbf{G} Cov(Z(B))\right) + \|\bar{\mathbf{g}}\|^2. \tag{5}$$

Once the above result is established, it is straightforward to show the following by observing the fact that the sample average is an unbiased estimator of the overall average, i.e., $\mathbb{E}_B[\mathcal{G}(B)] = \bar{\mathbf{g}}$.

**Corollary 1.** *Assume all the assumptions in Lemma 1 hold. Then for any random set $B$ with fixed size $|B|$, the following holds:*

$$\mathbb{E}_B\left[\|\mathcal{G}(B) - \bar{\mathbf{g}}\|^2\right] = \frac{1}{|B|^2} trace\left(\mathbf{G}^\top \mathbf{G} Cov(Z(B))\right) \tag{6}$$

Finally, we use the above results to prove the following which calculates the expected squared distance between the sample average and the overall average based on individual vectors and the overall average. The following result is critical because later we will see that Equation (7) connects the mini-batch stochastic gradient, the batch gradient, and individual gradients in Problem (P).

**Theorem 1.** *Assume all the assumptions in Lemma 1 hold. If elements of the random set $B$ are drawn uniformly and without replacement, then*

$$\mathbb{E}_B\left[\|\mathcal{G}(B) - \bar{\mathbf{g}}\|^2\right] = \frac{N - |B|}{|B|N(N-1)}\left(\sum_{i=1}^{N} \|\mathbf{g}^{(i)}\|_2^2 - N\|\bar{\mathbf{g}}\|^2\right) = \frac{N - |B|}{|B|N}\frac{1}{N-1}\sum_{i=1}^{N} \|\mathbf{g}^{(i)} - \bar{\mathbf{g}}\|_2^2. \tag{7}$$

### 4.2 STOCHASTIC RESULTS FOR HARD THRESHOLDING OPERATOR

The goal of this subsection is to show the random sequence $\left(f(\mathbf{x}^k)_{k\geq 1}\right)$ generated by the mini-batch SIHT algorithm converges with probability one. To show this we prove that the random sequence of the function value is a supermartingale sequence so the expected value of the function value sequence is decreasing. To achieve our goal, we prove the following lemma that provides an upper bound on the function value evaluated at a thresholded vector. Notice that the following result does not require the input be an updated vector by the gradient.

**Lemma 3.** *Let $f : \mathbb{R}^n \to \mathbb{R}$ be in $C^1$ and Ls-RSS. Then for a fixed $\mathbf{x} \in C_s$ with any $\mathcal{I}_s^{\mathbf{x}}$, any $0 < \gamma \leq \frac{1}{L_s}$, and any given vector $\mathbf{g} \in \mathbb{R}^n$, either of the following holds for any $\mathbf{y} \in H_s(\mathbf{x} - \gamma\mathbf{g})$ with any $\mathcal{I}_s^{\mathbf{y}}$:*

$$f(\mathbf{y}) \leq f(\mathbf{x}) - \frac{\gamma}{2}(1 - L_s\gamma)\|\mathbf{g}_{\mathcal{I}_s^{\mathbf{y}}}\|_2^2 - \frac{\gamma}{2}\|\mathbf{g}_{\mathcal{I}_s^{\mathbf{x}}}\|_2^2 + \gamma\langle\delta_{\mathcal{I}_s^{\mathbf{y}}}, \mathbf{g}_{\mathcal{I}_s^{\mathbf{y}}}\rangle + \gamma\langle\delta_{\mathcal{I}\setminus\mathcal{I}_s^{\mathbf{y}}}, \mathbf{x}_{\mathcal{I}\setminus\mathcal{I}_s^{\mathbf{y}}}\rangle \quad (8)$$

*where $\mathcal{I} = \mathcal{I}_s^{\mathbf{x}} \cup \mathcal{I}_s^{\mathbf{y}}$ and $\delta = \mathbf{g} - \nabla f(\mathbf{x})$.*

Observe that in the above lemma the vector $\mathbf{g}$ can be any vector in $\mathbb{R}^n$. It need not be the gradient nor the mini-batch gradient. However, in the following lemma we prove that if $\mathbf{g}$ is designated to be an unbiased stochastic approximation of the gradient at an arbitrary point, then the following result holds.

**Lemma 4.** *Let $f : \mathbb{R}^n \to \mathbb{R}$ be in $C^1$ and Ls-RSS. Assume $\mathbf{g}(\mathbf{x}, \omega)$ be an unbiased stochastic approximation of the gradient at $\mathbf{x} \in \mathbb{R}^n$ where $\omega \sim D$ for some distribution $D$, i.e., $\mathbb{E}_\omega[\mathbf{g}(\mathbf{x}, \omega)] = \nabla f(\mathbf{x})$. Then for a fixed $\mathbf{x} \in C_s$ with any $\mathcal{I}_s^{\mathbf{x}}$ and $0 < \gamma \leq \frac{1}{L_s}$, either of the following holds for any $\mathbf{y}(\omega) \in H_s(\mathbf{x} - \gamma\mathbf{g}(\mathbf{x}, \omega))$ with any $\mathcal{I}_s^{\mathbf{y}(\omega)}$:*

$$\mathbb{E}_\omega[f(\mathbb{Y}(\omega))] \leq f(\mathbf{x}) - \frac{\gamma}{2}(1 - L_s\gamma)\mathbb{E}_\omega[\|\mathbf{g}_{\mathcal{I}_s^{\mathbb{Y}(\omega)}}(\mathbf{x}, \omega)\|_2^2] - \frac{\gamma}{2}\|\nabla_{\mathcal{I}_s^{\mathbf{x}}}f(\mathbf{x})\|_2^2 + \gamma\mathbb{E}_\omega[\|\delta_{\mathcal{I}_s^{\mathbb{Y}(\omega)}}(\omega)\|_2^2] \quad (9)$$

*where $\mathcal{I}(\omega) = \mathcal{I}_s^{\mathbf{x}} \cup \mathcal{I}_s^{\mathbb{Y}(\omega)}$ and $\delta(\omega) = \mathbf{g}(\mathbf{x}, \omega) - \nabla f(\mathbf{x})$.*

The following Theorem is the climax of our technical results because it establishes a stochastic gradient descent property for the expectation of the function value. Later we will see how Inequality (11) is used in Theorem 3 to show the sequence of the function values generated by the mini-batch SIHT is a supermartingale sequence.

**Theorem 2.** *Let $f^{(i)} : \mathbb{R}^n \times \Xi \to \mathbb{R}$ be in $C^{1\ [1]}$ for $i = 1, \dots, N$ and $\Xi = \{\xi^{(1)}, \dots, \xi^{(N)}\}$ be a given set such that $f(\mathbf{x}, \Xi) = \frac{1}{N}\sum_{i=1}^N f^{(i)}(\mathbf{x}, \xi^{(i)})$ be an $L_s$-RSS function. Assume there exists a $c > 0\ [2]$ such that*

$$\mathbb{E}_{\mathcal{J}}\Big[\sum_{i=1}^N \|\nabla_{\mathcal{J}}f^{(i)}(\mathbf{x}, \xi^{(i)})\|_2^2\Big] \leq c\mathbb{E}_{\mathcal{J}}\Big[\|\nabla_{\mathcal{J}}f(\mathbf{x}, \Xi)\|_2^2\Big] \quad (10)$$

*for all $\mathbf{x} \in \mathbb{R}^n$ and any random index set $\mathcal{J} \subseteq \{1, \dots, n\}$ with $|\mathcal{J}| \leq s$. Let $\mathcal{G}(\mathbf{x}, \Xi, B) = \frac{1}{|B|}\sum_{i\in B}\nabla f^{(i)}(\mathbf{x}, \xi^{(i)})$ be the mini-batch stochastic gradient at any $\mathbf{x} \in \mathbb{R}^n$ where $B \subseteq \{1, \dots, N\}$ be a random set whose elements are drawn randomly and uniformly from $\{1, \dots, N\}$ without replacement and its size is $|B|$. For a fixed $0 < \gamma < \frac{1}{L_s}$, assume the size of $B$ is fixed such that $|B| \geq N/\left(1 + \frac{1-L_s\gamma}{1+L_s\gamma}\frac{N-1}{\frac{c}{N}-1}\right)$ and let $\zeta := \frac{N-|B|}{|B|(N-1)}$ for $N \geq 2$. Then for a fixed $\mathbf{x} \in C_s$ with any $\mathcal{I}_s^{\mathbf{x}}$ the following holds for any $\mathbb{Y}(B) \in H_s(\mathbf{x} - \gamma\mathbf{g}(\mathbf{x}, \Xi, B))$ with any $\mathcal{I}_s^{\mathbb{Y}(B)}$:*

$$\begin{aligned}
\mathbb{E}_B\Big[f(\mathbb{Y}(B), \Xi)\Big] \leq{}& f(\mathbf{x}, \Xi) - \frac{\gamma}{2}\|\nabla_{\mathcal{I}_s^{\mathbf{x}}}f(\mathbf{x})\|_2^2 \\
& - \frac{\gamma}{2}(1 + L_s\gamma)\zeta\Big(1 - \frac{c}{N} + \frac{1-L_s\gamma}{1+L_s\gamma}\frac{1}{\zeta}\Big)\mathbb{E}_{\mathcal{I}_s^{\mathbb{Y}(B)}}\Big[\|\nabla_{\mathcal{I}_s^{\mathbb{Y}(B)}}f(\mathbf{x}, \Xi)\|^2\Big]
\end{aligned} \quad (11)$$

*where $1 - \frac{c}{N} + \frac{1-L_s\gamma}{1+L_s\gamma}\frac{1}{\zeta} \geq 0$.*

---

[1]The class consisting of all differentiable functions whose derivative is continuous.

[2]In Remark 1, we explain why such a $c$ always exist for widespread objective functions in machine learning applications

A crucial assumption for proving the results in Theorem (11) is the assumption made in Inequality (10). In the following Claim we show that for a certain class of functions $c > 0$ always exists and it does not depend on the function. We will prove that for these special classes of functions the value of $c$ only depends on the data.

**Claim 2.** *Let the given set $\Xi$ in Problem (P) be defined such that $\Xi := \{\mathbf{V}_{1\bullet}, \ldots, \mathbf{V}_{N\bullet}\}$ where each $\mathbf{V}_{i\bullet}$ is the $i$-th row of a given matrix $\mathbf{V} \in \mathbb{R}^{N \times n}$. Then the objective function in Problem (P) can be defined as $f(\mathbf{x}, \Xi) := \frac{1}{N} \sum_{i=1}^{N} f^{(i)}(\mathbf{V}_{i\bullet}\mathbf{x})$ $f^{(i)} : \mathbb{R}^n \times \Xi \to \mathbb{R}$ and the following holds:*

$$\sum_{i=1}^{N} \|\nabla_{\mathcal{J}} f^{(i)}(\mathbf{V}_{i\bullet}\mathbf{x})\|_2^2 \leq \frac{N^2}{\sigma_{min}^2(\mathbf{V}\mathbf{I}_{\mathcal{J}\bullet}^\top \mathbf{I}_{\mathcal{J}\bullet}\mathbf{V}^\top)} \Big( \max_{r=1,\ldots,N} \big\{ \|(\mathbf{V}_{r\bullet}^\top)_{\mathcal{J}}\|_2^2 \big\} \Big) \|\nabla_{\mathcal{J}} f(\mathbf{x}, \mathbf{V})\|_2^2 \quad (12)$$

*where $\mathcal{J} \subseteq \{1, \ldots, n\}$ with $|\mathcal{J}| \leq s$, $\mathbf{I}_{\mathcal{J}\bullet} \in \mathbb{R}^{|\mathcal{J}| \times n}$ is a restriction of the Identity matrix whose rows are associated with indices in $\mathcal{J}$, $\mathbf{V}\mathbf{I}_{\mathcal{J}\bullet}^\top \mathbf{I}_{\mathcal{J}\bullet} = \sum_{i=1}^{|\mathcal{J}|} \mathbf{V}_{\bullet i}\mathbf{V}_{\bullet i}^\top$, $\sigma_{min}(\cdot)$ is the smallest singular value, $\mathbf{V}_{\bullet i}$ is the $i$-th column of $\mathbf{V}$, and $(\cdot)\mathcal{J}$ is a vector restricted to indices in $\mathcal{J}$.*

**Remark 1.** *The above claim shows that for a class of functions $f(\mathbf{x}, \Xi) := \frac{1}{N} \sum_{i=1}^{N} f^{(i)}(\mathbf{V}_{i\bullet}\mathbf{x})$ the constant $c > 0$ in Theorem 3 always exists and it does not depend on the value of $\mathbf{x}$ or its gradient whether it is batch (full) gradient or individual one. For an example of functions belonging to this class one can think of the mean square error loss used for linear regression as follows:*

$$f(\mathbf{x}, \mathbf{V}) = \frac{1}{N}\|\mathbf{V}\mathbf{x} - \mathbf{y}\|^2 = \frac{1}{N} \sum_{i=1}^{N} (\mathbf{V}_{i\bullet}\mathbf{x} - y_i)^2$$

*where $\mathbf{V} \in \mathbb{R}^{N \times n}$, $\mathbf{V}_{i\bullet}$ is the $i$-th row of $\mathbf{V}$, $\mathbf{x} \in \mathbb{R}^n$ is the optimization variable, and $\mathbf{y} \in \mathbb{R}^N$ is the target. Also, the logistic regression loss (binary cross entropy) is a function for which $c > 0$ in Inequality (12) always exists since it can be written as follows:*

$$f(\mathbf{x}, \mathbf{V}) = \frac{1}{N} \sum_{i=1}^{N} \Big( -y^{(i)}(\mathbf{V}_{i\bullet}\mathbf{x}) + \log\big(1 + e^{\mathbf{V}_{i\bullet}\mathbf{x}}\big) \Big)$$

*where $\mathbf{V} \in \mathbb{R}^{N \times n}$ whose last column is all one, $\mathbf{V}_{i\bullet}$ is the $i$-th row of $\mathbf{V}$, $\mathbb{R}^n \ni \mathbf{x} = [\mathbf{w}, b]^\top$ such that $\mathbf{w} \in \mathbb{R}^{n-1}$ and $b \in \mathbb{R}$ are the optimization variables, and $y^{(i)} \in \{0, 1\}$ for $i = 1, \ldots, N$.*

Now we can provide a result showing that by fixing a sparse point, one can use the stochastic mini-batch gradient with a fixed mini-batch size determined in Theorem 3 and decrease the function value in expectation.

**Theorem 3.** *Assume all the assumptions in Theorem 2 hold. Then for a fixed $\mathbf{x} \in C_s$ with any $\mathcal{I}_s^\mathbf{x}$ the following holds for any $\mathbb{Y}(B) \in H_s(\mathbf{x} - \gamma\mathcal{G}(\mathbf{x}, \Xi, B))$:*

$$\mathbb{E}_B\Big[ f(\mathbb{Y}(B), \Xi) \Big| \mathbf{x} \Big] \leq f(\mathbf{x}, \Xi) - \frac{\gamma}{2}\|\nabla_{\mathcal{I}_s^\mathbf{x}} f(\mathbf{x})\|_2^2. \tag{13}$$

The above result is the analogue result to (Damadi & Shen, 2022, Corollary 1).

**Theorem 4.** *Assume all the assumptions in Theorem 2 hold. Let $f$ be a bounded below differential function and $(\mathbb{X}^k | \mathbb{X}^{k-1})_{k \geq 0}$ be the stochastic IHT sequence. Then, $\Big( f(\mathbb{X}^k, \Xi, B) \Big| \mathbb{X}^k \Big)_{k \geq 1}$ is a supermartingale sequence and converges to a random variable $f^*$ with probability one.*

## 5 CONCLUSION

We showed the stochastic sequence generated by the mini-batch stochastic IHT is a supermartingale sequence converging with probability one. To show this result we used the stochastic gradient descent property that we derived where we utilized the property of the mini-batch stochastic gradient as the sample sum of a finite sum.

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
