# OpenReview forum: "Convergence of the mini-batch SIHT algorithm"
_ICLR.cc/2023/Conference — Submitted to ICLR 2023_

### Official Review · Reviewer_HcXm · 2022-10-20

**Confidence:** 3
**Correctness:** 1
**Technical Novelty And Significance:** 3
**Empirical Novelty And Significance:** Not applicable
**Recommendation:** 1

**Clarity, Quality, Novelty And Reproducibility:**

As described in the above section, due to the lack of details (proofs + unclear statement of assumptions in theoretical results) it is very hard to reproduce the results and the paper is very unclear.

I even find it hard to make a thorough assessment of the novelty and quality of the results.

**Strength And Weaknesses:**

Unfortunately, the manuscript at hand seems very incomplete. Thus, at its current state it is impossible to assess its contributions.

1. The current paper does not contain a single. Thus, I am not able to see whether some of those statements are correct..
2. Some of the assumptions for the theoretical statements are too vague and not clearly stated. (I will give some examples below.)
3. The current paper does not include any simulations. It would be great if the authors could include some numerical experiments to corroborate their theoretical results. This is maybe less important in comparison to the first two points but I also think that a complete submission should also contain numerical experiments.

Here are some more specific points:
1. Lemma 2: You say that B is a random set. What is the probability distribution? Where is it specified? In a similar vein, what is the probability that the Bernoulli random variable $z_i (B)$ is one? Are the  $z_i (B)$  i.i.d. random variables?
2. Lemma 3: What is meant by "either of the following holds". I only can see one statement.
3. Theorem 2: What is the distribution of the random set $\mathcal{J}$?
4. Theorem 4: How is this random variable $f^{\star}$ characterized?


Typos:
1. page 1, abstract: "...we do not assume the function to be (a) restricted strongly convex."
2. Lemma 4: "Assume $g(x,\omega)$ be (is) an..."
3. Theorem 2: "where $ B \subset  {1; ... ; N } $ be (is)"
4. Claim 2, definition of $f(x, \Xi)$: Is there an $f^{(i)}$ too much? Also should $f^{(i)} be $f$ instead?!
5. Claim 2: Should "$VI_J^T I_J=...$" read "$ VI_J^T I_J V^T=...$" instead?
6. Theorem 4: differential -> differentiable?!

**Summary Of The Paper:**

This paper studies the problem of minimising an objective function over all $s$-sparse vectors. In order to solve this non-convex problem, the authors study stochastic mini-batch of version of the well-known iterative hard thresholding algorithm.

This paper makes the following contributions:
1. It establishes a stochastic gradient descent property (Theorem 2).
2. This property is used to show that the objective values at each iteration form a super martingale, which is claimed to imply convergence.


**Summary Of The Review:**

Due to the shortcomings described above, the paper seems to be very incomplete and I cannot recommend acceptance.

---

### Official Review · Reviewer_jszD · 2022-10-25

**Confidence:** 5
**Clarity, Quality, Novelty And Reproducibility:** It is hard to evaluate as there is a …
**Correctness:** 1
**Technical Novelty And Significance:** 1
**Empirical Novelty And Significance:** 1
**Recommendation:** 1

**Strength And Weaknesses:**

1. A new lower bound is provided.
2. The designed method is proven to be supermartingale.

**Summary Of The Paper:**

The authors study the mini-batch Stochastic IHT (SIHT) algorithm for solving sparse optimization problems.  A new lower bound is found to help design the mini-batch size. The loss sequence generated by SIHT is proved to be a supermartingale sequence and converges with probability one.

**Summary Of The Review:**

The reviewer assumes the submission is incomplete work. There is a lack of proof in Theorems and lemmas. No experiments to demonstrate the designed method is better/worse than current methods.

---

### Official Review · Reviewer_mF9G · 2022-10-26

**Confidence:** 4
**Correctness:** 2
**Technical Novelty And Significance:** 2
**Empirical Novelty And Significance:** Not applicable
**Recommendation:** 3

**Clarity, Quality, Novelty And Reproducibility:**

On novelty, I find that at least one very related paper is missing from the discussion [1] (probably others too). This paper presents a convergence analysis of stochastic IHT for functions that have the restricted smooth and strongly convexity assumptions. By the way, the strong convexity assumption can be guaranteed by regularizing the loss function.

On reproducibility, some of the results like Lemmas 3-4 seem true, while e.g. Claim 2 is hard to verify. Again, since the proofs aren't included, this has a negative effect on reproducibility.

[1]
Ac/dc: Alternating compressed/decompressed training of deep neural networks
Peste, Alexandra and Iofinova, Eugenia and Vladu, Adrian and Alistarh, Dan

**Strength And Weaknesses:**

Strengths
- Advancing the theoretical understanding of stochastic IHT is an important task.
- The writing is generally easy to follow.

Weaknesses
- The literature review is quite limited.
- The proofs are missing, so it is not possible to fully verify the claimed results.
- In Claim 2, the minimum singular value should be 0, because the matrix is low-rank. If the authors define minimum singular value differently, it is hard to know because the proof is not included.
- In Claim 2, an upper bound is derived for c (10). Based on this, $c$ seems to be $\geq N^2$. When I plug that back into Theorem 2, I get a batch size of roughly $N - \Theta(1)$. This does not seem like a great choice for a batch size, which typically should be much less than $N$. It would be great if the authors could comment on whether I am missing something here.
- There are no experiments, which could shed light for example on what is the value of c in practice.

**Summary Of The Paper:**

The authors study mini-batch stochastic iterative hard thresholding (IHT). IHT is a well known algorithm for sparse optimization with extensive literature. IHT is essentially $\ell_2$-projected gradient descent on the set of s-sparse vectors. Given the ML-wide focus on stochastic optimization algorithms, it is natural to study the guarantees of stochastic IHT.

The main result of the paper is a martingale convergence result that claims stochastic IHT converges to the optimal function value, although no convergence rate is given.

**Summary Of The Review:**

In summary, while I like the problem of minibatch stochastic IHT and the approach, I find that there are significant issues with novelty that should be resolved in the literature review section, as well as my other concerns outlined above. I encourage the authors to continue working on the manuscript since I think this is an important subject.

---

### Decision · Program_Chairs · 2023-01-20

**Decision:**

Reject

**Justification For Why Not Higher Score:**

For the reasons mentioned above

**Justification For Why Not Lower Score:**

N/A

**Metareview: Summary, Strengths And Weaknesses:**

- Summary:

The authors study mini-batch stochastic iterative hard thresholding (IHT).
IHT is a well known algorithm for sparse optimization with extensive literature.
Given the ML-wide focus on stochastic optimization algorithms, it is natural to study the guarantees of stochastic IHT.

- Strengths
Findings of the paper include:
1. The paper provides theoretical results for stochastic IHT;
2. The paper proves arguments as a supermartingale;
3. A new lower bound is provided.

- Weaknesses
1. All reviewers have found the paper rather incomplete.
2. Reviewers found that the literature review is quite limited.
3. There are no experiments, which could shed light on how the proposed algorithm + theory connects with practice.
4. Reviewers state that it is hard to evaluate the lemmas + theorems as the proofs are missing
5. Some of the assumptions for the theoretical statements are too vague and not clearly stated.

- What would be missing:
1. Better connection with existing literature
2. Inclusion of proofs
3. Extensive results on the proposed algorithm